# Agave Fructans as a Carbon Source to Develop a Postbiotic-Based Strategy for the Prophylaxis and Treatment of *Helicobacter pylori* Infection

**DOI:** 10.3390/ijms262211119

**Published:** 2025-11-17

**Authors:** Enrique A. Sanhueza-Carrera, Natalia C. Hernández-Delgado, Carolina Romo-González, César Castro-De la Mora, Claudia Mendoza-Camacho, Cynthia Fernández-Lainez, Gabriel López-Velázquez

**Affiliations:** 1Laboratorio de Biomoléculas y Salud Infantil, Instituto Nacional de Pediatría, Ciudad de México 04530, Mexico; natalia.hernandez-delgado@outlook.com; 2Posgrado en Ciencias Biológicas, Universidad Nacional Autónoma de México, Ciudad de México 04510, Mexico; 3Secretaría de Ciencia, Humanidades, Tecnología e Innovación (SECIHTI), Ciudad de México 03940, Mexico; 4Laboratorio de Bacteriología Experimental, Instituto Nacional de Pediatría, Ciudad de México 04530, Mexico; cromog@pediatria.gob.mx; 5Layan Biotic Solutions, Guadalajara 44670, Mexico; cesarcastro@bggroup.mx (C.C.-D.l.M.); claudia.m@bggroup.mx (C.M.-C.); 6Laboratorio de Errores Innatos del Metabolismo y Tamiz, Instituto Nacional de Pediatría, Ciudad de México 04530, Mexico; lainezcynthia@ciencias.unam.mx

**Keywords:** agave, fructans, *Helicobacter*, postbiotics, biofilm

## Abstract

*Helicobacter pylori* is a Gram-negative bacterium that inhabits the gastric mucosa and infects over 50% of the global population, predominantly in developing countries. The organism causes chronic gastritis and is associated with gastric carcinoma. Traditional antibiotic treatment promotes intestinal dysbiosis and antimicrobial resistance. In this context, postbiotics—the metabolic end products of probiotics—have been shown to be powerful antimicrobial alternatives. The excretion/secretion (E/S) products and exopolysaccharides (EPSs) of lactic acid bacteria (LAB) have been found to exhibit inhibitory activity against pathogens. EPSs are complex sugar polymers involved in biofilm formation and stress resistance, and their activity varies with culture conditions. Most notably, no digestible carbohydrates, such as those present in agave-derived Graminan-Type fructans (GTFs), are effective carbon sources for LAB, which, in turn, affects their metabolic end products. In this study, the E/S products and EPSs of the INP_MX_001 LAB strain were assayed for antimicrobial and antibiofilm activity after growth with three structurally different GTFs. Results indicated potent inhibition of *H. pylori* survival and biofilm formation in vitro. Our results confirm the promise of using LAB-derived postbiotics, particularly those produced with GTFs, as a novel, non-antibiotic means of combating *H. pylori* colonization and infection.

## 1. Introduction

*Helicobacter pylori* is a pathogenic, microaerophilic, Gram-negative bacterium that colonizes the gastric mucosa due to its motility and resistance to gastric acidity. This pathogen has a worldwide prevalence of infection greater than 50%, primarily in developing countries [1,2]. Between 2015 and 2022, the prevalence in adults was 43.9%, whereas in children and adolescents, it was 35.1% [3]. *H. pylori* is the causative agent of a gastroduodenitis characterized by chronic inflammation, long-lasting destruction of the mucosal barrier, which can lead to gastric cancer. Likewise, the infection that this pathogen causes has also been associated with various extragastric diseases, including colorectal cancer [4,5].

The standard treatment for *H. pylori* infection consists of proton pump inhibitors (PPIs), clarithromycin, and either amoxicillin or metronidazole, administered for 10 to 14 days. However, antibiotics can alter the intestinal microbiota, cause various side effects, and contribute to antibiotic resistance [5,6]. Consequently, there is a need to explore novel therapeutic strategies that are both safer and more effective, either alone or in combination with other therapies.

An alternative to treating this infection is the use of probiotics, prebiotics, postbiotics, and synbiotics [7]. Probiotics, including Lactic Acid Bacteria (LAB), have emerged as promising candidates for treating *H. pylori* infection [8]. The addition of probiotics to the therapeutic regimen may modify the gastrointestinal microbiota and reduce medication-related adverse effects [9]. For instance, it has been demonstrated that *Limosilactibacillus reuteri* DSM 17648 can bind to *H. pylori* in the gastric milieu, forming copolymers that facilitate its elimination by interfering with the pathogen’s adhesion to the gastric mucosa [10]. Additionally, in cell cultures and animal models, *Lactobacillus acidophilus* NCFM, *L. acidophilus* La-14, *Lactiplantibacillus plantarum* Lp-115, *Lacticaseibacillus paracasei* Lpc-37, *Lacticaseibacillus rhamnosus* Lr-32, and *L. rhamnosus* GG strains reduce the adhesion and consequently decrease inflammation by *H. pylori* [11].

Furthermore, the postbiotics—by products of probiotic metabolism, such as cell lysates, proteins, short-chain fatty acids, and exopolysaccharides (EPSs)—have been investigated as antimicrobial agents [12]. Thus, antimicrobial activity against commensal pathogens in the excretion/secretion (E/S) products, such as bacteriocins, organic acids, and exopolysaccharides (EPSs) of various LAB, has been demonstrated [12,13].

The EPSs play a key role in biofilm formation, cell adhesion, and protection against adverse environmental conditions. For example, the EPSs from *Lactiplantibacillus plajomi* PW-7 disrupt the cell membranes of pathogenic bacteria, having a minimum inhibitory concentration of 50 mg/mL [5]. EPSs are defined as branched and repeating units of sugars or sugar derivatives, which are long-chain, high molecular weight polymers produced by various microorganisms, including LAB [12]. It is worth noting that the structures of the EPSs are strongly influenced by culture conditions, such as temperature, pH, and carbon source, which, in turn, influence the functions they may perform [14,15].

An important source of carbon is non-digestible carbohydrates (NDCs). A significant group of NDCs is fructans, which are fructose-based carbohydrate polymers that differ in their glycosidic linkages [16]. Fructans with a branched structure contain a mixture of fructofuranosyl units linked by β (2-1) and β (2-6) glycosidic bonds [17], and they are also known as graminan-type fructans (GTFs), which can be extracted from plants such as agave. These GTFs have demonstrated their utility as a carbon source for LAB [14].

Research has shown that certain LAB strains, when cultured with various GTFs extracted from agave and used as carbon sources, exhibit distinct immunomodulatory effects through their E/S products and EPs [14].

We aimed to identify a potential strategy for managing *H. pylori* infections using postbiotics. To achieve this, we cultured the INP_MX_001 LAB strain with three structurally distinct agave GTFs as carbon sources, then isolated the E/S products and EPSs. We then evaluated these LAB postbiotics in vitro for their effects on the survival and biofilm formation of three highly pathogenic *H. pylori* strains.

## 2. Results

### 2.1. Prophylactic and Therapeutic Effects of LAB Strain E/S Products Against H. pylori

We wondered whether the E/S products of the INP_MX_001 LAB strain have prophylactic and therapeutic effects on the growth and biofilm formation of *H. pylori*. To achieve this, we used 10^7^–10^8^ CFU/mL of *H. pylori* for prophylaxis assays [18,19,20]. For therapeutic assays, we used the previously reported *H. pylori* count for patients with active chronic gastritis (10^9^ CFU/mL) [15,21].

Figure 1 illustrates the prophylactic effects of the E/S products from the LAB strain grown in the presence of GTF A, GTF B, and GTF C as carbon sources. It is observed that the E/S products from LAB grown in different GTFs inhibited the growth by 45 to 70% in *H. pylori* strains J99 (Figure 1A, left panel), 26695 (Figure 1B, left panel), and 43504 (Figure 1C, left panel), being more effective than the control Lp-115 strain, with statistically significant differences (*p* < 0.0001) compared with the untreated control. However, it was less effective against the 43504 strains, showing lower activity than obtained with the Lp-115 strain.

Additionally, E/S products from LAB grown in the different GTFs inhibited biofilm formation by 0.61 ± 0.18 to 0.74 ± 0.13-fold, being more effective against the J99 strain than the untreated control (Figure 1A, right panel) and to a lesser extent against the 43504 strains by 0.53 ± 0.09 to 0.67 ± 0.07-fold (Figure 1B, right panel). However, they did not inhibit biofilm formation by the 26695 strains (Figure 1C, right panel).

The therapeutic effect of the E/S products from LAB grown with different GTFs affected the survival of J99, 26695, and 43504 strains, with a range from 0.51 ± 0.02 to 0.63 ± 0.009-fold change, being more effective than the control Lp-115 strain with a range from 0.61 ± 0.01 to 0.83 ± 0.03-fold change (Figure 2, left panels). The weakest effect was observed for E/S products obtained from the INP_MX_001 strain grown with glucose (*p* < 0.0001).

On the other hand, the therapeutic effect of E/S products on biofilm formation was more substantial against the 43504 strain with 0.59 ± 0.04 to 0.76 ± 0.04-fold (Figure 2C, right panel), followed by the impact on the J99 strain (Figure 2A, right panel), where the use of GTF B (0.77 ± 0.11-fold change) and GTF C (0.80 ± 0.14-fold change) was the most effective (*p* < 0.0001). Inhibition of biofilm formation was less effective with the 26695 strains (Figure 2B, right panel).

### 2.2. Prophylactic Effects Against H. pylori of EPSs Extracted from the INP_MX_001 LAB Strain Using the Different GTFs as Carbon Sources

We evaluated the EPSs obtained from the INP-MX-001 LAB strain as postbiotics, using the infective doses of *H. pylori* for prophylactic and therapeutic assays as described above. It is worth noting that the effects were dependent on the carbon source used for growth. The prophylactic approach of EPSs from the INP-MX-001 LAB strain was assessed at 1 and 2 mg/mL. In this range, an effect on the growth of *H. pylori* was detected (Appendix A); therefore, these concentrations were chosen for further experiments. EPSs were able to significantly inhibit (*p* < 0.0001) the growth of the three *H. pylori* strains J99 (Figure 3A, left panel), 26695 (Figure 3B, left panel), and 43054 (Figure 3C, left panel) with a range of 0.24 ± 0.004 to 0.58 ± 0.01-fold change. Additionally, all the assayed EPSs from the INP-MX-001 LAB strain showed more effective effects than those obtained from the strain Lp-115 for the 26695 and 43504 strains. For the J99 strain, EPSs derived from the INP-MX-001 LAB strain with GTF A as a carbon source demonstrated greater efficacy than those from Lp-115 (Figure 3A).

However, at 2 mg/mL, all EPSs from the INP-MX-001 LAB strain showed greater growth inhibition than those from Lp-115 (Figure 4, left panels). At this concentration, the highest inhibitory activity was observed in the 43504 strain, being more active with a range from 0.21 to 0.26-fold than the EPSs of Lp-115 (Figure 4C, left panel).

We also investigated whether the two EPS concentrations mentioned above inhibit biofilm formation, thereby potentially increasing the prophylactic effects already observed. Although highly effective at 1 mg/mL, the therapeutic approach against biofilm formation in the J99 strain did not show a dose-dependent effect (Figure 3A, right panel vs. Figure 4A, right panel). Even at 2 mg/mL, the inhibition of biofilm formation in the J99 strain was less effective, especially with EPSs obtained from the INP-MX-001 LAB strain grown in GTF B (Figure 4A, right panel). At 1 mg/mL, only GTF B (0.81 ± 0.14-fold) and GTF C (0.75 ± 0.23-fold) were significantly effective against biofilm formation of the 26695 strains (Figure 3B, right panel). The EPSs from the three GTFs as carbon sources were significantly effective (0.64 ± 0.10 to 0.73 ± 0.03-fold) against biofilm formation in 43504 strains (Figure 3C, right panel). With 2 mg/mL, EPSs exhibited the major effect on biofilm inhibition against 26695 (0.66 ± 0.10 to 0.68 ± 0.03-fold change) and 43504 strains (0.54 ± 0.09 to 0.63 ± 0.03-fold change) (*p* < 0.0001; Figure 4B,C, right panels).

### 2.3. Therapeutic Effects of 1 and 2 mg/mL of EPSs Extracted from the INP_MX_001 LAB Strain Using the Different GTFs as Carbon Sources Against H. pylori

The following in vitro results demonstrated the potential effectiveness of these EPSs as postbiotics in treating infections caused by the J99, 26695, and 43504 strains. This series of assays demonstrated carbon sources and dose-dependent effects on the growth of *H. pylori* strains, particularly at 2 mg/mL EPSs, with values ranging from 0.31 ± 0.003 to 0.62 ± 0.03-fold (*p* < 0.0001), exhibiting dose-dependent behavior, except for the EPSs from GTF B (Figure 5B and Figure 6B, left panels).

Furthermore, in the inhibition of *H. pylori* biofilm formation, the EPSs from GTF A, GTF B, and GTF C showed a significant dose-dependent effect against the 26695 and 43504 strains, with inhibition ranging from 0.49 ± 0.08 to 0.80 ± 0.08-fold (*p* < 0.0001) (Figure 5B,C and Figure 6B,C, left panels). For strain J99, the 2 mg/mL concentration did not enhance biofilm inhibition; instead, it reduced effectiveness with EPS from GTF B. (Figure 5A and Figure 6A, right panels).

## 3. Discussion

The use of postbiotics is considered Generally Recognized as Safe (GRAS) [22,23]. Postbiotics exhibit beneficial functional properties for health, including the inhibition of gastrointestinal pathogens [24,25,26]. Since the properties of postbiotics vary notably among species, strains, and culture conditions [21,27,28], we wondered if the postbiotics produced by our LAB strain grown with three structurally different GTFs as a carbon source would have anti-*H. pylori* effects. Previous studies have demonstrated a carbon-source dependence in the immunomodulatory effects of E/S products and EPSs from LAB strains [14]. To the best of our knowledge, the present study is the first report demonstrating the anti-*H. pylori* activity of postbiotics derived from a LAB strain grown with different GTFs from agave plants.

The *H. pylori* strains studied here were selected based on the presence of pathogenic islands and their antibiotic resistance profiles (Appendix A) [28]. These virulence factors are key in determining targeted therapies for treating this pathogen [29].

Considering that *H. pylori* is strongly associated with chronic gastritis, peptic ulcers, and even gastric cancer, it becomes crucial to investigate both prophylactic and therapeutic conditions, as this knowledge can guide prevention strategies and improve long-term clinical outcomes [30]. Moreover, the relationship with biofilm formation is relevant to both approaches, since prophylactic assays aim to prevent initial adhesion and the establishment of biofilms, while therapeutic assays seek to reduce or eliminate already formed biofilms, thereby promoting the effective eradication of the pathogen [31].

The prophylactic assay of E/S products exhibited significant growth-inhibitory activity (between 65% and 70%) against the three *H. pylori* strains, even surpassing that of the reference *L. plantarum* Lp-115 strain. The E/S products from the INP_MX_001 LAB strain showed the same effectiveness as the J99 and 26695 strains, regardless of the carbon source. The GTF B, which exhibits the same proportion of long and short fructan chains, was the most efficient in inducing the production of E/S with anti-*H. pylori* 43504 strain activity. Chen et al. obtained similar results, assaying the E/S products of *L. gasseri* and *L. plantarum* against *H. pylori* SS1, achieving 60% growth inhibition [32]. Unlike the *H. pylori* strains studied herein, SS1 primarily colonizes rodents and lacks a functional cag pathogenicity island (cag-PAI). This suggests that we have identified a promising postbiotic system with potential utility in preventing this type of disease in humans.

The therapeutic assays demonstrated that the E/S products significantly reduced the bacterial load of the *H. pylori* strains studied at initial densities greater than 10^9^ CFU/mL. The 40% to 55% clearance rate we obtained is relevant, considering that the number of bacteria we studied is comparable to those observed in chronic infections or persistent mucosal colonization. This is the first study to use such a high concentration of *H. pylori* and to evaluate postbiotics as a potential treatment. Similar approaches have been performed, but only using complete bacterial cells of *Lactiplantibacillus plantarum* LN66, *Lactiplantibacillus plantarum* B7, and *Lactobacillus acidophilus* LB [33,34,35]. Although studying smaller amounts of pathogens, others have proposed that the use of E/S products from LABs affects *H. pylori* motility and movement [36], inhibiting growth and adhesion [37], and suppressing virulence and survival genes while modulating the immune response against the pathogen [38].

On the other hand, the EPSs showed a stronger prophylactic growth-inhibitory activity (between 62% and 80%) than the E/S products against the *H. pylori* strains. Except for the EPSs from the LAB grown in GTF B, the prophylactic assays with EPSs showed a dose-dependent effect with 40% to 70% growth-inhibitory activity at 2 mg/mL. All the assayed EPSs from the INP_MX_001 LAB strain inhibited the growth and survival of *H. pylori* more efficiently than the *L. plantarum* Lp-115 strain. Our EPSs were even more efficient than those of *L. plantarum* previously assayed in animal models [11].

Of importance are the carbon-source- and dose-dependent effects exhibited by EPSs in both types of treatment. This reinforces results from other studies on probiotics, highlighting the importance of metabolite type, dosage, and administration timing in optimizing effectiveness [39,40].

The effectiveness of postbiotics as natural products for coping with pathogens is reinforced by the results reported herein. Regarding the role of EPSs in such anti-pathogenic processes, other studies with EPSs produced by *Lactobacillus* sp. PW-7 demonstrated a growth-inhibitory effect and damage to the cellular integrity of *H. pylori* in a dose-dependent manner [5]. In another study, Srinivash and collaborators tested the effect of the EPSs produced by *Lactococcus hircilactis* CH4 and *Lactobacillus delbrueckii* at concentrations ranging from 25 to 100 μg/mL [41]. Bamisi and collaborators evaluated the antimicrobial activity of secondary metabolites from probiotics against *Salmonella enterica*, *Shigella dysenteriae*, and *Escherichia coli*, obtaining inhibition halos ranging from 6.8 mm to 11.7 mm [42]. Additionally, the use of EPSs from the *Streptococcus thermophilus* CRL1190 strain decreased *H. pylori* adhesion to 65.9% and reduced the inflammatory response in cellular models [43]. On this line, Garcia-Castillo and colleagues demonstrated the in vivo ability of *L. fermentum* UCO-979C to increase resistance against *H. pylori* infection, and that its EPSs are partially responsible for its immunomodulatory effect, as they reduce IL-8 levels and enhance IL-10 levels [44].

Biofilms are of great importance because they facilitate the persistence of *H. pylori* in the stomach, protecting the bacteria from antibiotics and the immune system, and contributing to resistance to conventional treatments. Therefore, targeting their formation is crucial to prevent colonization and infection [45].

Previous studies have shown that the E/S products of LAB can inhibit biofilm formation by 50–90% in *Listeria monocytogenes*, *Escherichia coli* O157:H7, and ciprofloxacin-resistant *Salmonella typhimurium* [43]. These effects have been evaluated using standardized inocula (10^6^–10^8^ CFU/mL), ensuring the development of robust, reproducible biofilms comparable to those observed in chronic infections or persistent mucosal colonization [46]. Additionally, these E/S products have been shown to reduce the adhesion of enteric pathogens, such as *E. coli* O157:H7 and *Vibrio cholerae*, by more than 80% in human cell line models (HT-29, Caco-2, HCT-116), without affecting host cell viability or the composition of the commensal microbiota [47,48,49]. These inhibitory effects are associated with the downregulation of key genes involved in extracellular matrix formation, such as *icaA*, *hilA*, *crl*, and *csgA* [47,50].

Based on the aforementioned, we evaluated the impact of E/S products from the INP_MX_001 LAB strain on *H. pylori* biofilm formation using the well-characterized J99, 26695, and 43504 strains, as well as a standardized inoculum of these pathogens. Except for the 26695 strains, our results support the potential effectiveness of using the assayed E/S products in inhibiting pathogen biofilm formation (Figure 1, right panels). Moreover, this biofilm-forming-inhibitory effect may be useful for preventing pathogen colonization, as observed with the J99 and 43504 strains in the prophylactic assays. For instance, the strongest prophylactic effect of the E/S products against biofilm formation was with the *H. pylori* 43504 strain. On the other hand, under therapeutic conditions, the biofilm formation-inhibitory effect was weaker. Our findings support the hypothesis that postbiotics are more effective at interfering with the early stages of adhesion and colonization than with established biofilm structures [51,52]. This aligns with previous studies demonstrating the greater prophylactic efficacy of LAB E/S products against pathogens such as *L. monocytogenes*, *Staphylococcus aureus*, and *Salmonella enterica* [47,51,53]. Although specific *H. pylori* molecular pathways were not assessed in this study, it is reasonable to hypothesize that the observed reduction in biomass is linked to interference with the expression of genes such as *ureA*, *vacA*, *cagA*, or quorum-sensing regulators involved in the transition from cells in the supernatant to biofilm formation [54,55]; however, further studies are needed to confirm it. Given that biofilm formation in *H. pylori* is associated with chronic persistence, immune evasion, and antibiotic resistance, postbiotic products emerge as a promising alternative, especially in the context of increasing resistance to clarithromycin and metronidazole, with rates exceeding 20–30% in various global regions.

By evaluating the effect of the INP_MX_001 LAB strain-derived EPSs on *H. pylori* biofilms, it was found that under prophylactic treatment at 1 mg/mL, the EPSs significantly reduced biofilm formation. This finding aligns with the results of Giordani et al. (2023), who reported that EPSs from *Lactobacillus gasseri* and *L. crispatus* inhibited the growth of urogenital pathogen biofilms and promoted the formation of beneficial *Lactobacillaceae* biofilms in a dose-dependent fashion [56]. Similarly, *Streptococcus thermophilus* CRL1190 exhibited strong prophylactic potential against *H. pylori*, enhancing probiotic adhesion and reducing pathogen colonization through EPS-dependent mechanisms [43]. When we increased the EPSs concentration to 2 mg/mL, under prophylactic conditions, the biofilm formation-inhibitory effect was enhanced. This latter finding is consistent with García-Castillo et al. (2020), who demonstrated that EPSs from *Lactobacillus fermentum* UCO-979C enhanced local immune responses and increased host resistance to *H. pylori* infection by promoting beneficial biofilm formation and stimulating anti-inflammatory cytokines, such as IL-10 [44,46,47,57]. Likewise, Rezaei et al. reported that *Lactiplantibacillus plantarum* and *Lacticaseibacillus rhamnosus* exhibited strong preventive effects by reducing pathogenic biofilm formation in early-stage competition models [58].

Despite the EPSs’ biofilm formation-inhibitory effect being weaker under therapeutic conditions, at 1 mg/mL, the effect persisted. This finding is consistent with Xu et al. (2020), who reported that EPSs from *Lactobacillus casei* NA-2 can adhere to pathogens such as *E. coli*, *S. aureus*, and *S. typhimurium*, thereby favoring partial disruption of mature biofilms [59,60,61]. Additionally, Wang et al. (2019) reported that EPSs from *Lactobacillus fermentum* S1 exerted therapeutic effects by altering the extracellular matrix of established biofilms through both structural and antioxidant mechanisms [59].

Finally, therapeutic conditions at 2 mg/mL EPSs showed a more pronounced anti-biofilm effect than at the lower concentration, although it was still less effective than prophylactic treatments. Similar results were reported by Song et al. (2020), who found that EPSs from *Lactobacillus plantarum* 12 disrupted mature *Shigella flexneri* biofilms by reducing cellular adhesion and lowering the minimum concentration required for biofilm eradication [62]. Altogether, these findings reinforce that LAB-derived EPSs serve not only as preventive agents but also possess therapeutic potential for interfering with established biofilms, being more effective during the early stages of colonization.

The results of this study highlight the biomedical potential of postbiotics derived from Lactic Acid Bacteria grown with agave GTFs as carbon sources. This strategy represents a significant innovation by proposing the use of postbiotics as therapeutic and preventive agents that can complement existing clinical treatments, thereby improving the efficacy and safety in managing infectious diseases.

From a biotechnological perspective, the production of differential postbiotics using agave GTFs as a carbon source for LAB strains enables the generation of bioactive metabolites with specific and targeted antimicrobial properties [61]. The advantage of using postbiotics is that viability is not an issue, unlike probiotics. Postbiotics are highly stable, have a longer shelf life, and, given their safety profile, represent an opportunity for development and pharmaceutical applications [12,63]. Furthermore, since they consist of defined mixtures of metabolites, standardization and quality control are more feasible, facilitating their industrial scalability [63,64]. The E/S products and exopolysaccharides studied herein may have health benefits against pathogens, but they may also be used for other applications, such as functional ingredients or additives in fermented foods.

Each *H. pylori* strain possesses intrinsic characteristics such as the expression of a distinct set of virulence genes, such as *cag*A, *vac*A, *ure*A/B, or *bab*A, that differentially modulate adhesion, urease production, and the host inflammatory response, thereby influencing the effectiveness of the postbiotics studied here against bacterial growth and biofilm formation. We hypothesize that the factors underlying the observed effects may be strain dependent.

Other factors that could be responsible for the differential effect observed are (a) the metabolic variations of the LAB, which were carbon-source-dependent (i.e., E/S), and (b) differences in the structure and composition of the produced EPSs. Such metabolic variations directly affect the synthesis of short-chain fatty acids (SCFAs), bacteriocins, hydrogen peroxide, and other antimicrobial metabolites, which can interfere with *H. pylori* growth and adhesion through mechanisms yet to be elucidated.

## 4. Materials and Methods

### 4.1. LAB Strains and Culture Conditions

We used an axenic LAB strain, a Gram-positive bacterium, catalase- and oxidase-negative, non-sporulated, of porcine origin, previously characterized by our research group for its immunomodulatory properties [14]. The reference code INP_MX_001 identified this LAB, and the species was determined to be *Enterococcus lactis*. The strain was preserved at −70 °C in a sterile solution of MRS medium, Man Rogosa Sharpe (MRS) (Difco™, Franklin Lakes, NJ, USA), supplemented with 50% (*w*/*v*) sterile glycerol. For its activation, a 100 µL aliquot of the strain preserved at −70 °C was thawed and inoculated into 10 mL of sterile MRS broth in test tubes. The tubes were incubated at 37 °C and 5% CO_2_ for 48 h. As anti-*H. pylori* reference strain, *Lactiplantibacillus plantarum* Lp-115, was used. This strain has previously been reported to inhibit the strains *H. pylori* P12 and *H. pylori* SS1, as tested in a human gastric adenocarcinoma cell line [11]. The storage, activation method, and culture conditions of this reference strain were identical to those used for INP_MX_001 LAB strain.

### 4.2. H. pylori Strains and Culture Conditions

The *H. pylori* strains J99 (ATCC 700824), 26695 (ATCC 700392), and NCTC 11637 (ATCC 43504), preserved in Brucella broth (Difco™, Franklin Lakes, NJ, USA) with 20% glycerol, were thawed and cultured on blood agar base plates (BD Bioxon™, Mexico City, Mexico) supplemented with 10% sheep blood (Merck Millipore, Darmstadt, Germany). The plates were incubated at 37 °C with 9% CO_2_ for 5 days. Subsequently, growth was assessed for purity by performing a Gram stain and biochemical tests for urease, oxidase, and catalase. All strains yielded positive results. An inoculum of each strain was then suspended in saline solution to prepare a McFarland 3 suspension for the inhibition assays. These *H. pylori* strains were selected for this study based on their characteristics, including antimicrobial resistance profiles and virulence factors (Appendix A) [65,66,67,68,69,70,71,72,73,74,75,76].

### 4.3. Graminan-Type Fructans

The GTFs used as carbon sources for the growth of the LAB strain were GTF A, Fiber Fructans™ (Guadalajara, Mexico); GTF B, Flora Advantage™ (Guadalajara, Mexico); and GTF C, Fiber Balance™ (Guadalajara, Mexico), all isolated from Mexican agave plants (Industrializadora de Fructanos Tierra Blanca™, Guadalajara, Mexico). The chemical and structural characteristics of these GTFs have been previously described [14].

### 4.4. Cultures of LAB in MRS–Graminan-Type Fructans Modified Media

The INP_MX_001 LAB strain was cultured in a modified MRS medium without carbohydrates (cfMRS) [18]. The formulation contained: 7 g of meat extract (Merck Millipore Darmstadt, Germany), 7 g of peptone (Merck Millipore Darmstadt, Germany), 5 g of yeast extract (Merck Millipore Darmstadt, Germany), 3 g of sodium acetate × 3 H_2_O (Sigma–Aldrich, St. Louis, MO, USA), 2 g of K_2_HPO_4_ (Sigma–Aldrich, St. Louis, MO, USA), 1.2 g of NH_4_Cl (Sigma–Aldrich, St. Louis, MO, USA), 0.2 g of MgSO_4_ × 7 H_2_O (Sigma–Aldrich, St. Louis, MO, USA), and 0.06 g of MnCl_2_ × 4 H_2_O (Sigma–Aldrich, St. Louis, MO, USA). The pH was adjusted to 6.5 with 1 N HCl (Merck Millipore, Darmstadt, Germany) [14]. As the primary carbon source for fermentation, three media based on the above formulation were supplemented with 20 g/L of GTF A (MRS–A), GTF B (MRS–B), and GTF C (MRS–C), plus a standard MRS broth in which the main carbon source was glucose (20 g/L) (Merck Millipore, Darmstadt, Germany). The INP_MX_001 LAB strain was cultured separately in the three MRS media with different GTFs at 37 °C and 5% CO_2_ and in standard MRS-glucose medium for 48 h, while the reference strain *L. plantarum* Lp-115 with anti-*H. pylori* activity was cultured in MRS-glucose medium for 48 h, until both LAB strains under study reached an OD600nm growth of 1.0, which corresponds to a growth of approximately 1.6 × 10^9^ CFU/mL. After these cultures, the cells were centrifuged to obtain cell-free supernatants, which were then filtered through 0.22 µm filters. Aseptic conditions of the culture media were confirmed by inoculating aliquots into sterile MRS broth and agar, with no colony formation or microbial growth detected. These supernatants were identified as GTF A, GTF B, and GTF C, based on the GTFs used in fermentation and glucose, which was only present in the fermentation from MRS-glucose medium. From the pellet obtained from this centrifugation, EPSs were isolated as described below.

### 4.5. Extraction and Quantification of EPSs from LAB Pellets Under Study

With modifications, the method described by Ferrari et al. was used to extract EPSs from the bacterial pellet [77]. Bacterial cells obtained by centrifugation from previous fermentations were treated with 15 mL of 2 M NaOH (Merck Millipore, Darmstadt, Germany) overnight at room temperature. After centrifugation, the supernatant was separated from the bacterial pellet, and two volumes of 95% cold ethanol were added to the supernatant. The resulting mixture was incubated at −20 °C overnight. The pellets were collected by centrifugation, resuspended in sterile distilled water, and dialyzed (3.5 kDa MWCO) in 1 L of distilled water for 24 h, with two water changes per day. Total EPS quantification was carried out using the phenol-sulfuric acid method [78], using GTFs from Fiber Fructans™ as a standard for the calibration curve (10 μg/mL to 100 μg/mL).

### 4.6. In Vitro Inhibition of H. pylori Strains by EPSs and E/S Products of LAB

The effect on *H. pylori* was assessed using an in vitro prophylactic assay that determined the pathogen’s survival in the presence of the E/S products and the EPSs obtained from the LAB. The infectious dose of 10^7^–10^8^ CFU/mL, which is equivalent to the so-called McFarland 2, was used for each *H. pylori* strain [75]. This treatment was incubated for 72 h under culture conditions for *H. pylori* growth.

The effect of E/S products and EPSs of LAB on reducing the bacterial load of *H. pylori* was investigated using an in vitro therapeutic assay with a high infective load of approximately 10^9^ CFU/mL, equivalent to McFarland 4 [75].

To adjust both McFarland 2 (10^7^–10^8^ CFU/mL) and McFarland 4 (10^9^ CFU/mL) units in *H. pylori* strains, the bacteria were collected using a sterile swab directly from the surface of the incubated blood agar plate, and these bacteria were used to inoculate a sterile flask with 30 mL of *Brucella* medium supplemented with 10% fetal bovine serum and 10 mM urea. The bacterial suspension of each *H. pylori* strain was divided in half to perform assays with both the E/S products and the EPSs from the LAB under study.

To carry out each of the exposure assays with the E/S products from LAB and EPSs, 100 µL containing 10^7^–10^8^ CFU/mL (prophylactic) or 1 × 10^9^ CFU (therapeutic) of each *H. pylori* strain was used (Figure 7). The aforementioned aliquots of each *H. pylori* strain were mixed with 100 µL containing E/S products resulting from the INP_MX_001 LAB strain fermentation using GTF A, GTF B, GTF C, or MRS-glucose (coming from 1.6 × 10^9^ CFU/mL of LAB), or with the EPSs obtained from these fermentations. INP_MX_001 LAB strain EPSs were assayed at 1 mg/mL and 2 mg/mL. Additionally, as a reference assay, the prophylactic and therapeutic experiments were performed with the probiotic reference strain *L. plantarum* Lp-115, only with the E/S products from MRS-glucose fermentation (using 1.6 × 10^9^ CFU/mL) and with the EPSs obtained from this same fermentation (1 and 2 mg/mL). To reach the CFU number for the prophylactic or therapeutic assays, we used spectrophotometric measurements at OD600 nm. 96-well plates were incubated under microaerophilic conditions (10% CO_2_, 37 °C) for 72 h. At the end of incubations, we evaluated the growth inhibition effects at OD600 nm. The absorbance values for the different treatments were compared with those of the untreated control. Six technical replicates were performed, with each technical replicate carried out in triplicate.

The three *H. pylori* strains were mixed with E/S products and EPSs (1 and 2 mg/mL) obtained from fermentations of the INP_MX_001 isolate, in which GTF A, GTF B, and GTF C from agave were used as the sole carbon source for the LAB. From the *L. plantarum* Lp-115 strain, E/S products and EPSs were obtained when grown in MRS-glucose medium. These postbiotics were included as a positive control for *H. pylori* inhibition. The experimental workflow diagram was designed using Graphviz Online (https://dreampuf.github.io/GraphvizOnline/, accessed on 1 November, 2025).

### 4.7. In Vitro Inhibition of H. pylori Strains’ Biofilm Formation by EPSs and E/S Products of the INP_MX_001 LAB Strain

From the cultured plates resulting from the above-mentioned assays, the effect of the *H. pylori* strains’ biofilm inhibition was evaluated using the crystal violet staining method [18]. Briefly, the supernatant from each well was removed, and the wells were washed three times with sterile physiological saline solution (0.90% NaCl, *w*/*v*). In the last wash, the plates were inverted to remove non-adherent bacteria. The remaining cells were fixed on the plate with 200 μL of 99% methanol for 15 min. Then, the methanol was discarded, and the plates were left to dry in a biosafety cabinet. The adherent cells were stained with 200 μL of 2% crystal violet for 5 min. Afterwards, the crystal violet and excess were discarded with sterile distilled water washes. After drying the plate, the cells were resuspended in 200 μL of an alcohol-acetone mixture. For each treatment, OD was measured at 570 nm in a spectrophotometer. Six technical replicates were performed, with each technical replicate carried out in triplicate.

### 4.8. Statistical Analysis

Data normality was evaluated using the Shapiro-Wilk test. For normally distributed data, a one-way ANOVA with Dunnett’s multiple-comparison test was used. If the data did not follow a normal distribution, the Mann-Whitney or Friedman test was used, followed by Dunn’s test. The results are presented as mean ± standard deviation (SD) or median with interquartile range (IQR), as appropriate. A *p* < 0.05 was considered significant (* *p* < 0.05; ** *p* < 0.01; *** *p* < 0.001; **** *p* < 0.0001). At least six independent experiments, each with three technical replicates, were performed. The analysis was performed using GraphPad Prism (version 10.4.1).

## 5. Conclusions

The findings in the present study support the development of a novel strategy for creating postbiotics, such as E/S products and EPSs, from LAB strains grown with agave GTFs as carbon sources. Such a strategy demonstrated its effectiveness in producing anti-*Helicobacter* postbiotics, particularly in preventing colonization and infection by highly pathogenic *H. pylori* strains.

## Figures and Tables

**Figure 1 ijms-26-11119-f001:**
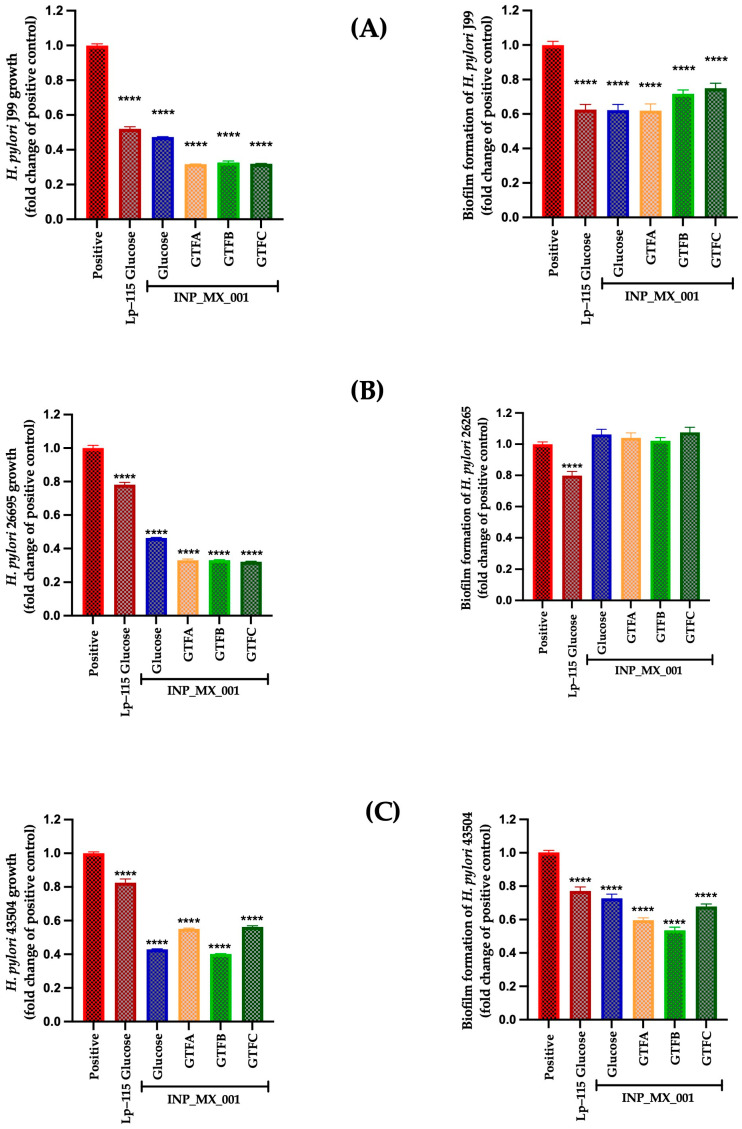
Prophylactic approach with E/S products from INP_MX_001 LAB strain against *H. pylori.* Bacterial growth (**left**) and biofilm formation (**right**) of (**A**) J99 (ATCC 700824), (**B**) 26695 (ATCC 700392), and (**C**) 43504 (ATCC 43504) strains. The results are plotted as the mean ± Std. deviation; a *p* < 0.05 was considered significant (**** *p* < 0.0001).

**Figure 2 ijms-26-11119-f002:**
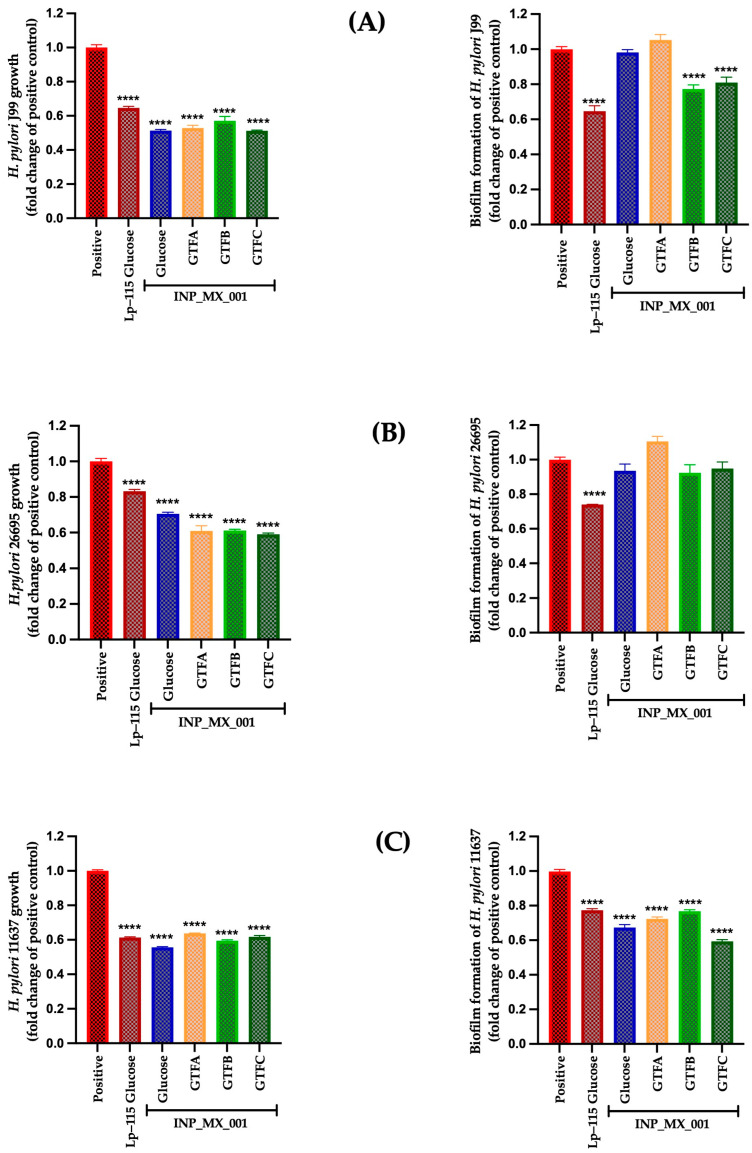
Therapeutic treatment with E/S products from the INP_MX_001 LAB strain against *H. pylori.* Bacterial growth (**left**) and Biofilm formation (**right**). (**A**) J99 (ATCC 700824) strain, (**B**) 26695 (ATCC 700392) strain, and (**C**) 43504 (ATCC 43504) strain. The results are plotted as the mean ± Std. deviation; a *p* < 0.05 was considered significant (**** *p* < 0.0001).

**Figure 3 ijms-26-11119-f003:**
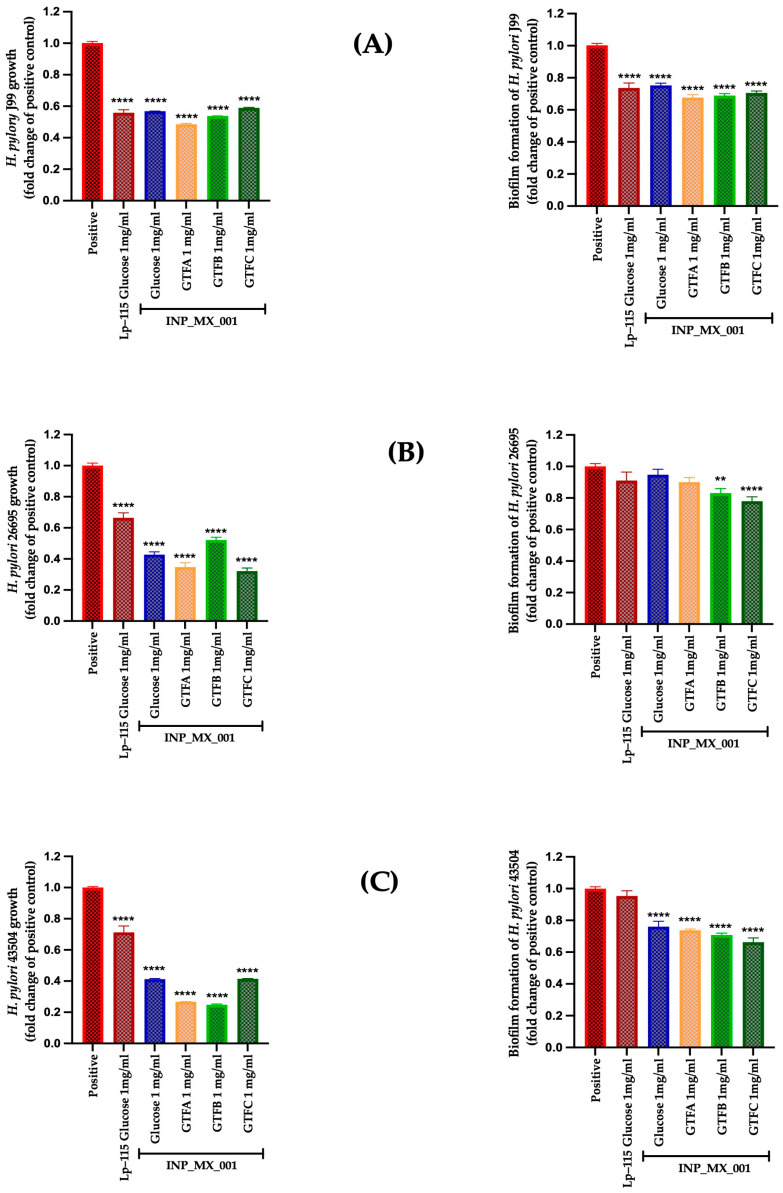
Prophylactic approach with 1 mg/mL of EPSs extracted from the INP_MX_001 LAB strain using the different GTFs as carbon sources against *H. pylori.* Bacterial growth (**left**) and Biofilm formation (**right**) using (**A**) J99 (ATCC 700824) strain, (**B**) 26695 (ATCC 700392) strain, and (**C**) 43504 (ATCC 43504) strain. The results are plotted as the mean ± Std. deviation; a *p* < 0.05 was considered significant (** *p* < 0.01; **** *p* < 0.0001).

**Figure 4 ijms-26-11119-f004:**
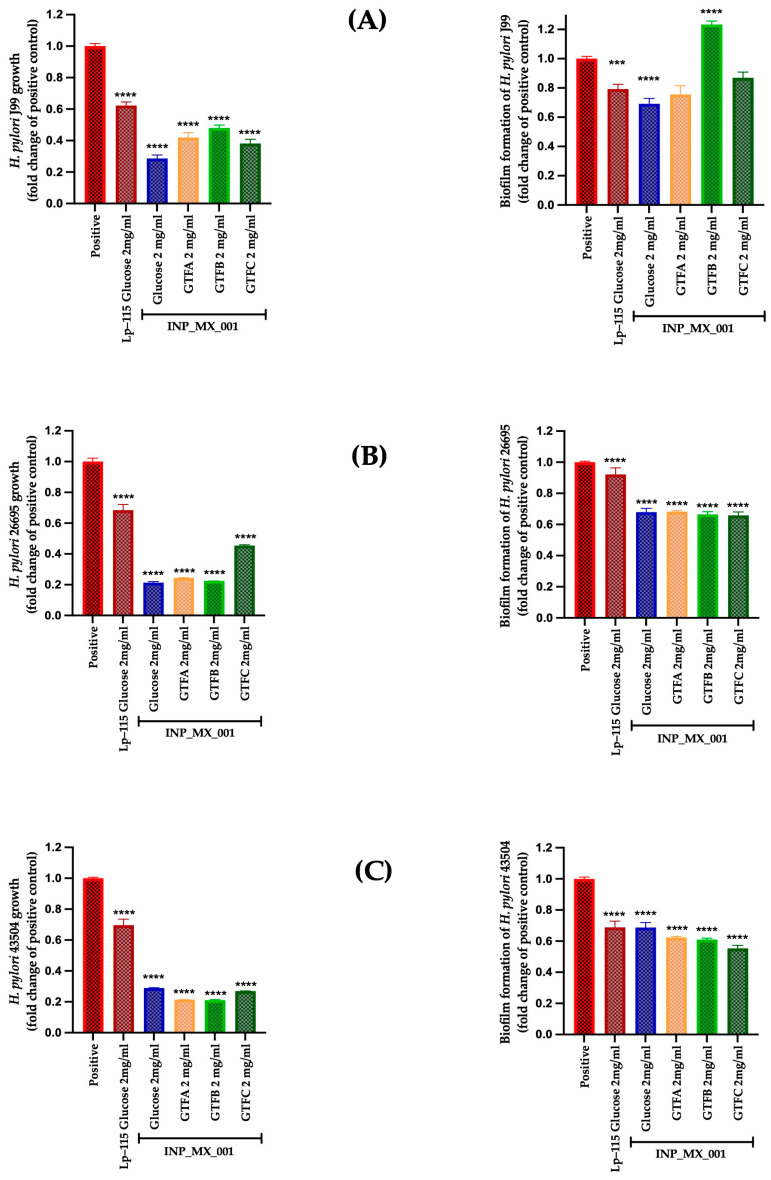
Prophylactic approach with 2 mg/mL of EPSs extracted from the INP_MX_001 LAB strain using the different GTFs as carbon sources against *H. pylori.* Bacterial growth (**left**) and Biofilm formation (**right**) using (**A**) J99 (ATCC 700824) strain, (**B**) 26695 (ATCC 700392) strain, and (**C**) 43504 (ATCC 43504) strain. The results are plotted as the mean ± Std. deviation; a *p* < 0.05 was considered significant (*** *p* < 0.001; **** *p* < 0.0001).

**Figure 5 ijms-26-11119-f005:**
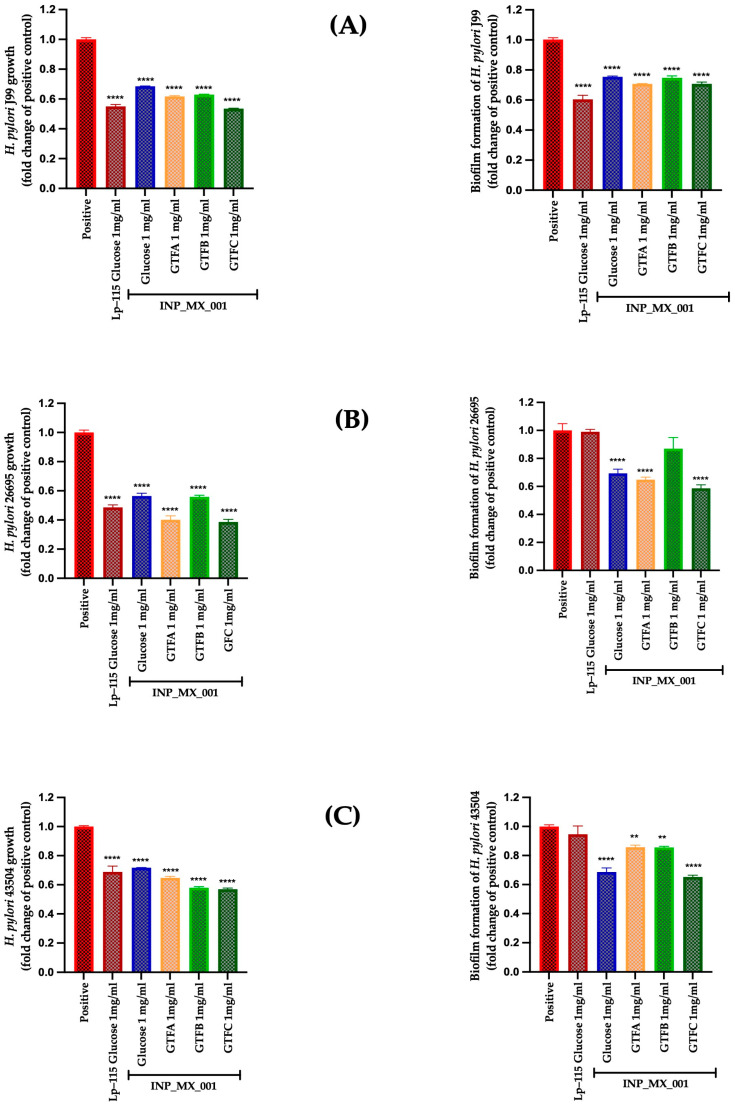
Therapeutic treatment with 1 mg/mL of EPSs extracted from the INP_MX_001 LAB strain using the different GTFs as carbon sources against *H. pylori*. Bacterial growth (**left**) and Biofilm formation (**right**) using (**A**) J99 (ATCC 700824) strain, (**B**) 26695 (ATCC 700392) strain, and (**C**) 43504 (ATCC 43504) strain. The results are plotted as the mean ± Std. deviation; a *p* < 0.05 was considered significant (** *p* < 0.01; **** *p* < 0.0001).

**Figure 6 ijms-26-11119-f006:**
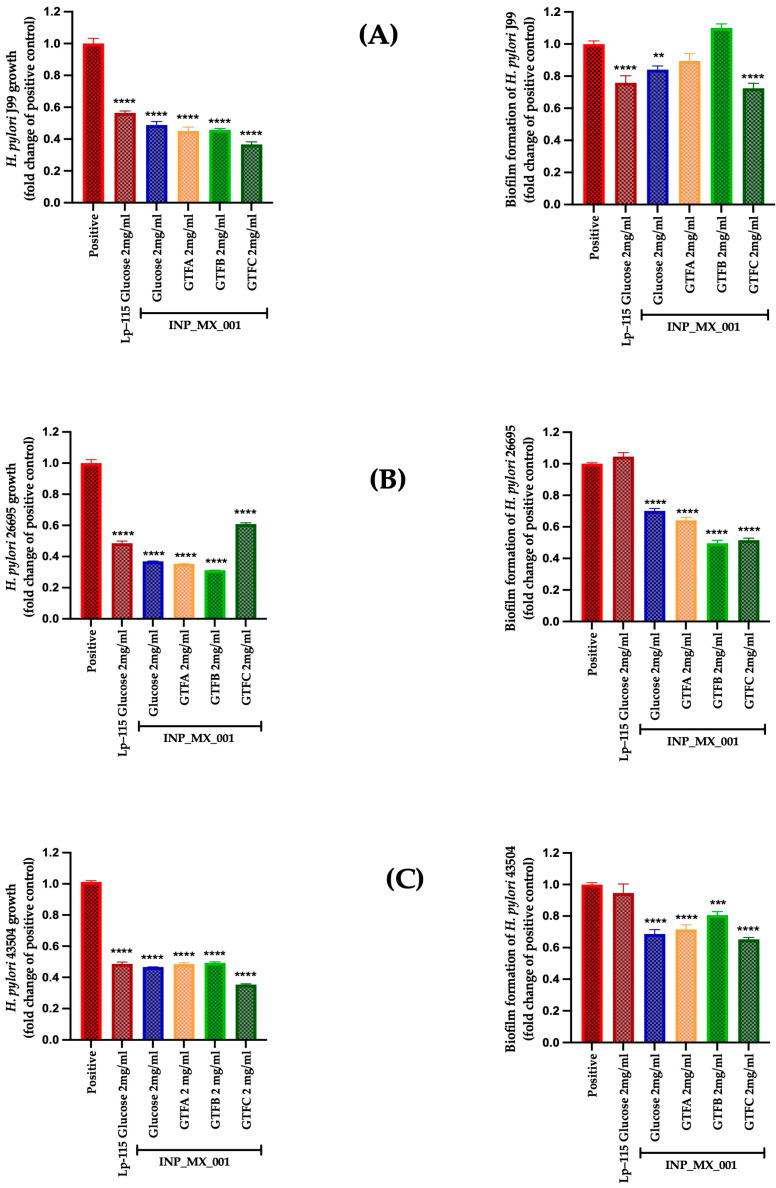
Therapeutic treatment with 2 mg/mL of EPSs extracted from the INP_MX_001 LAB strain using the different GTFs as carbon sources against *H. pylori*. Bacterial growth (**left**) and biofilm formation (**right**) using (**A**) J99 (ATCC 700824) strain, (**B**) 26695 (ATCC 700392) strain, and (**C**) NCTC 11637 (ATCC 43504) strain. The results are plotted as the mean ± Std. deviation; a *p* < 0.05 was considered significant (** *p* < 0.01; *** *p* < 0.001; **** *p* < 0.0001).

**Figure 7 ijms-26-11119-f007:**
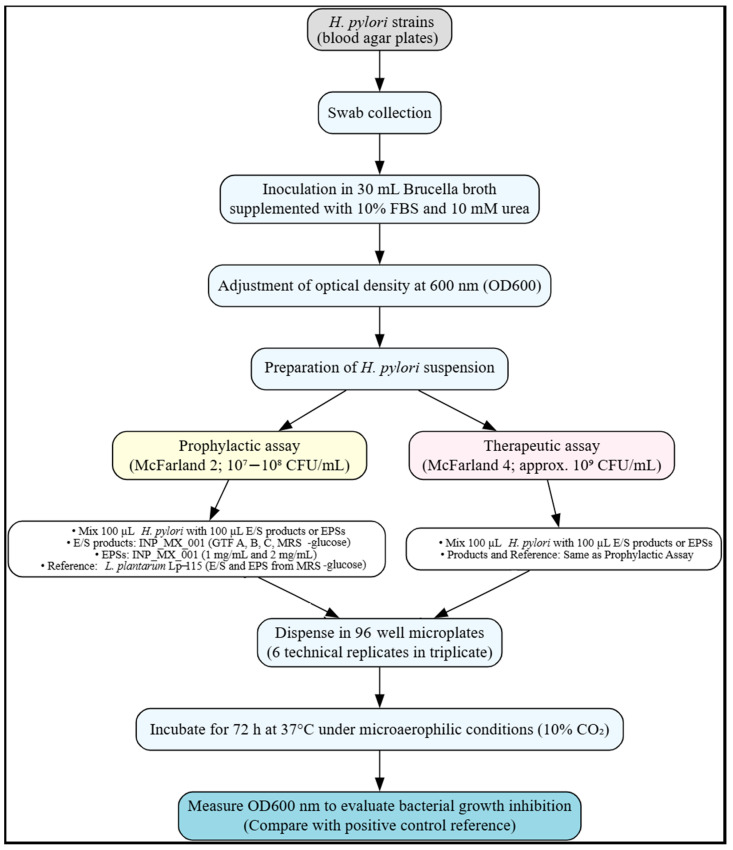
Schematic representation of the in vitro prophylactic and therapeutic assays against *Helicobacter pylori*.

## Data Availability

All the data used in the current study are available from the corresponding authors upon reasonable request.

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
