# Peer review of "Agave Fructans as a Carbon Source to Develop a Postbiotic-Based Strategy for the Prophylaxis and Treatment of Helicobacter pylori Infection"

_ijms, 2025, doi:10.3390/ijms262211119_

Round 1
Reviewer 1 Report
Comments and Suggestions for Authors
Helicobacter pylori will always be an important gastric pathogen to combat and postbiotics could be part of the solution in many ways or situations.
Line 29: I suggest to remove "for example" from the sentence.
Line 90: agave - I suggest writing that word in lowercase letters. Please, check the whole manuscript.
Line 101: Is it a Bacillus strain? Scientific names should be written in italics and capital first letter. Please check the whole manuscript. Or by “bacillus”, did you mean “rod shaped”?
Line 102: If the strain is a Bacillus species, it must be sporulated. Please, check the identification of the strain.
Maybe the characteristics of the strains used in the experiments should be presented in a table.
Line 144: MRS broth? Glucose is already in MRS medium. Did you use a different concentration of glucose?
Cell free supernatants: Information of the culture of supernatants in MRS agar should be added to verify the absence of bacteria.
Line 158: did you used a specific temperature incubation?
Line 168: McFarland 2 does not correspond to that concentration. Please verify the information. Maybe it is necessary to add or clarify other information.
Lines 180 – 182 is not clear. Please revise the sentence.
Maybe a flowchart should be added to show the experiments.
Line 182: “the later” what?
In Figures: the distance of the bar indicating the strain INP_MX_001 is confusing, perhaps it should cover less or not cover the control strain.
The idea or concept of the therapeutic and prophylactic assays should be explained/discussed regarding to the relationship with the biofilm formation.
Maybe a table with the resumed characteristics of the Helicobacter pylori strains used in the experiments should be added (isolation matrix, pathogenicity islands, virulence genes…).
Please, use the new nomenclature for Lactobacillus (check the whole manuscript). If you are referring to the genus Lactobacillus, it is better to use the concept of the family Lactobacillaceae.
Comments on the Quality of English Language
Some sentences are not well written or seems to be literally translated from spanish.
Author Response
Please the attachment, thank you.

Reviewer 2 Report
Comments and Suggestions for Authors
- How would a proven-effective prophylaxis or treatment to H. pylori work on the assay in Figure 1? Please considering add a positive control to the experiment. On the other hand, maybe clarify the effect of the reference Lp-115 strain, how effective is that one supposed to be?
- Is 1mg/ml & 2mg/ml tested in figure 3/4 physiologically relevant? Please clarify
- What is the differential content for the EPSs extracted from your LAB strain using different GTFs as carbon source? Please elaborate.
- How to explain the differential effect on the 3 different strains? Please elaborate and at least provide some potential hypothesis
Author Response
Please see the attachment, thank you.

Round 2
Reviewer 1 Report
Comments and Suggestions for Authors
The manuscript now shows the importance and complexity of research on Helicobacter pylori, and its relevance to the development of therapeutic alternatives.
Line 458: McFarland 2 is equivalent to the concentration of 6 x 108, not 107.
Reviewer 2 Report
Comments and Suggestions for Authors
I think it is ok to publish, they corrected some of the language issue and supplemented some discussion part for the differential response of different strains to E/S product.